METHODS

# Statistical integration of multi-omics and drug screening data from cell lines

**Said el Bouhaddani**[1][*], **Matthias Höllerhage**[2], **Hae-Won Uh**[1], **Claudia Moebius**[3], **Marc Bickle**[4], **Günter Höglinger**[2,5,6,7‡], **Jeanine Houwing-Duistermaat**[1,8‡]

**1** Dept. Data science & Biostatistics, UMC Utrecht, Utrecht, Netherlands, **2** Department of Neurology, Hannover Medical School, Hannover, Germany, **3** Max Planck Institute of Molecular Cell Biology and Genetics, Dresden, Germany, **4** Roche Institute for Translational Bioengineering, Basel, Switzerland, **5** Department of Neurology, Ludwig-Maximilians-Universität, Munich, Germany, **6** German Center for Neurodegenerative Diseases, Munich, Germany, **7** Munich Cluster for Systems Neurology (SyNergy), Munich, Germany, **8** Dept. of Mathematics, Radboud University, Nijmegen, Netherlands

☯ These authors contributed equally to this work.
‡ GH and JHD also contributed equally to this work.
* sbouhad2@umcutrecht.nl

**Data Availability Statement:** Software is available online at github.com/selbouhaddani/ MultiOmicsCompWorkflow. Transcriptome and

## Abstract

Data integration methods are used to obtain a unified summary of multiple datasets. For multi-modal data, we propose a computational workflow to jointly analyze datasets from cell lines. The workflow comprises a novel probabilistic data integration method, named POPLS-DA, for multi-omics data.

The workflow is motivated by a study on synucleinopathies where transcriptomics, proteomics, and drug screening data are measured in affected LUHMES cell lines and controls. The aim is to highlight potentially druggable pathways and genes involved in synucleinopathies. First, POPLS-DA is used to prioritize genes and proteins that best distinguish cases and controls. For these genes, an integrated interaction network is constructed where the drug screen data is incorporated to highlight druggable genes and pathways in the network. Finally, functional enrichment analyses are performed to identify clusters of synaptic and lysosome-related genes and proteins targeted by the protective drugs. POPLS-DA is compared to other single- and multi-omics approaches.

We found that *HSPA5*, a member of the heat shock protein 70 family, was one of the most targeted genes by the validated drugs, in particular by AT1-blockers. *HSPA5* and AT1-blockers have been previously linked to *α*-synuclein pathology and Parkinson's disease, showing the relevance of our findings.

Our computational workflow identified new directions for therapeutic targets for synucleinopathies. POPLS-DA provided a larger interpretable gene set than other single- and multi-omic approaches. An implementation based on R and markdown is freely available online.

## Author summary

We present a computational workflow that combines the analysis of different types of data measured in cell line studies with non-overlapping samples. We apply the workflow to

proteome data are published (https://doi.org/10.3389/fneur.2022.787059) and can be found at: Sequencing data at NCBI GEO, accession no: GSE191302 (https://www.ncbi.nlm.nih.gov/geo/query/acc.cgi?acc=GSE191302); Proteomics data at PRIDE, accession no: PXD028322 (https://www.ebi.ac.uk/pride/archive/projects/PXD028322). The raw FDA drug validation screening data are available upon request at the System Administrator of the Dept. of Neurology, MH Hannover (neurologie.sekretariat@mh-hannover.de). The aggregated drug validation data used in the workflow is given in the manuscript (Table 1).

**Funding:** All authors were funded by ERA-Net E-Rare JTC 2018 (MSA-omics) [40- 44000-98-2006/90030376507]. JD was supported by European Union's Horizon 2020 (IMforFUTURE) [721815] and the EU funded Cost action DYNALIFE [CA21169]. GH was funded by the Deutsche Forschungsgemeinschaft (DFG, German Research Foundation) under Germany's Excellence Strategy within the framework of the Munich Cluster for Systems Neurology (EXC 2145 SyNergy) [390857198] and within the Hannover Cluster RESIST (EXC 2155) [39087428], DFG [HO2402/18-1] MSAomics, the Bavarian Ministry for Education, Culture, Science and Art (ForIPS) [8810001412], Niedersächsisches Ministerium für Wissenschaft und Kunst [ZN3440.TP]: REBIRTH Forschungszentrum für translationale regenerative Medizin, VolkswagenStiftung (Niedersächsisches Vorab), Petermax-Müller Foundation (Etiology and Therapy of Synucleinopathies and Tauopathies). MH and GH received funding from ParkinsonFonds Deutschland (Hypothesis-free compound screening in a new human neuronal model of wild type alpha-synuclein-induced cell death). The funders had no role in study design, data collection and analysis, decision to publish, or preparation of the manuscript.

**Competing interests:** The authors have declared that no competing interests exist.

measurements of gene expression, protein abundances, and a screening of a wide range of FDA-approved drugs. These different types of data are obtained from LUHMES brain cells and jointly analyzed to discover new treatment options in synucleinopathies, such as Parkinson's disease. Our workflow includes a new probabilistic method, named POPLS-DA. POPLS-DA combines the analysis of the genes and proteins to pinpoint a set of relevant genes and proteins that can distinguish affected and non-affected cells. Compared to other approaches, POPLS-DA found a larger set of genes relevant to the disease. Further, we constructed a network that connects the relevant genes and proteins that interact with each other. We incorporate the drug screening data to highlight which part of the network is relevant to the disease and druggable. Through additional analysis of the functionality, we discovered that the genes and proteins that are targeted by protective drugs share relevant properties, namely they are synaptic and lysosome-related genes. Notably, we found that specific types of drugs, namely AT1-blockers such as Telmisartan, are protective and target the network of relevant genes and proteins. These drugs are approved by the FDA and readily available to further investigate their potential in treating synucleinopathies. We further found that a gene named HSPA5, a member of the heat shock protein 70 family, is highly targeted by the protective drugs. This gene has been linked to Parkinson's disease in previous scientific literature. Our computational workflow and the implementation in R and markdown are freely available online.

This is a *PLOS Computational Biology* Methods paper.

## Introduction

Nowadays, studies often include measurements of several omics datasets as well as other types of data. Particularly cell line studies can generate datasets such as genomics, transcriptomics, and proteomics in a reproducible and standardized experimental system. Experiments using cell lines have also been set up to measure the efficacy of FDA-approved drugs in rare diseases and identify potentially viable treatments. How to summarize and visualize these different datasets is an ongoing research topic. A joint analysis of all data is most efficient but also challenging since it might need a model describing the relationship between the datasets. Another way is to integrate datasets by using functional information about the top features of the datasets. This paper is motivated by a cell line study with proteomics, transcriptomics, and drug compound screening data. These data have been analyzed separately, but this did not yield an overarching overview of involved genes. New computational tools are needed to obtain such an overview of relationships between the omics data, and their interplay with drugs in the context of MSA. Here, we present a workflow to analyze the omics data jointly and integrate these with the drug compounds screening data using functional information.

A common approach to analyzing multiple omics data is to take the intersection of significant results of single dataset analyses. For such analyses, univariate t-tests or multivariate regression-based approaches are typically used [1]. For regression approaches, meta-analysis can also be used to combine results of single omics data analysis [2]. More advanced approaches involve converting the measurements per omics dataset to ranks and aggregating these ranks to a consensus list of top ranked features [3], or jointly modeling all multi-omics data using, e.g., MINT [4]. MINT is an adaptation of partial least squares discriminant analysis (PLS-DA) [5] to a multi-group approach. This method treats the omics datasets as subgroups

with respect to the total sample size and estimates linear combinations of features that maximally discriminate between cases and controls across all subgroups. An advantage of single dataset modeling is that it is less affected by heterogeneity across omics data, while the joint modeling approach MINT does not take into account this heterogeneity. It has been shown that modeling omics-specific variation might yield better results when analyzing heterogeneous datasets [6, 7]. Since transcriptomics and proteomics data are different (e.g. technical platform and scale), including data-specific components in the model is expected to be beneficial. We, therefore, propose a new approach, probabilistic OPLS-DA (POPLS-DA), which is an adaption of PO2PLS [8] for the analysis of heterogeneous multi-omics data from overlapping samples, to non-overlapping cell lines.

Concerning drug screen data, results are often integrated by identifying gene targets of validated drugs from databases and intersecting these with differentially expressed genes from other experiments [9]. In this paper, we will use a similar approach. Specifically, gene targets of significant drug compounds are integrated into the functional network of the top POPLS-DA genes using a 'direct neighbor' approach, where POPLS-DA genes that interact with the drug targets are highlighted.

In our motivating study, transcriptomic and proteomic data are measured in LUHMES cells. LUHMES cells are derived from human embryonic dopaminergic cells [10] and are used to investigate the neurobiological processes underlying neurodegeneration, specifically synucleinopathies. Synucleinopathies are neurodegenerative disorders where a pathological aggregation of $\alpha$-synuclein is present. The most common synucleinopathy is Parkinson's disease (PD) and multiple system atrophy (MSA). It is assumed that aggregated forms of $\alpha$-synuclein are toxic, leading to cell death and neurodegeneration. Integrative multi-omics approaches can provide insights into the biological processes, molecular functions, and interactions underlying the diseases. The transcriptomics and proteomics data were measured in cells overexpressing $\alpha$-synuclein or GFP as a control [1]. Furthermore, to investigate whether existing drugs can be the basis for a novel therapy against $\alpha$-synuclein-induced cell death, LUHMES cells were used to screen 1600 FDA-approved drugs. The primary screening was performed in triplicates in two different concentrations (3 $\mu$M and 10 $\mu$M) and revealed 53 drugs that reduced cell death in LUHMES cells overexpressing $\alpha$-synuclein in at least one of the screening runs [11]. Here, we propose to integrate the drug compound screening data with the top features from the omics data analysis by using functional information on the genes targeted by the drugs and the interactome. Thus we aim to perform an integrated analysis of the three datasets to obtain a holistic overview of gene deregulation in $\alpha$-synuclein overexpressing cells and to obtain a unified list of the most relevant features (genes and drug compounds) [12]. Note that such a list can comprise features that were not found with single dataset analyses, because of a lack of statistical evidence when analyzed separately.

Our contribution is firstly a novel data integration workflow comprising a joint analysis of the omics data and an integration approach using functional information to combine the top omic features and the drug compound screening results. Secondly, a new probabilistic method named POPLS-DA for joint analysis of omics datasets in cell lines is developed. Thirdly application of the workflow including POPLS-DA to transcriptomics, proteomics, and drug compound screening datasets in LUMES cell lines. We perform dose-response testing of all 53 compounds to validate the protective efficacy of the drug compounds. Then, we analyze the three datasets using our novel integrated computational workflow. We apply POPLS-DA to integrate the two omics datasets and identify relevant genes and proteins that distinguished the two experimental conditions ($\alpha$-synuclein versus control protein overexpression). Based on these genes and proteins, a protein-protein interaction (PPI) network was constructed and the genes/proteins that are targeted by the drugs were identified. This approach highlighted

genes in the network that are associated with $\alpha$-synuclein overexpression and targeted by validated drugs, and therefore are potential targets for a novel therapy for synucleinopathies.

## Results

We first present the results of the drug screen validation. Next, we show the results of our computational workflow (depicted in Fig 1) applied to the multi-omics and drug screening data from the LUHMES cells. Enrichment analyses are presented. Finally, we compare our novel multi-omic integration approach POPLS-DA with existing single- and multi-omics integration approaches.

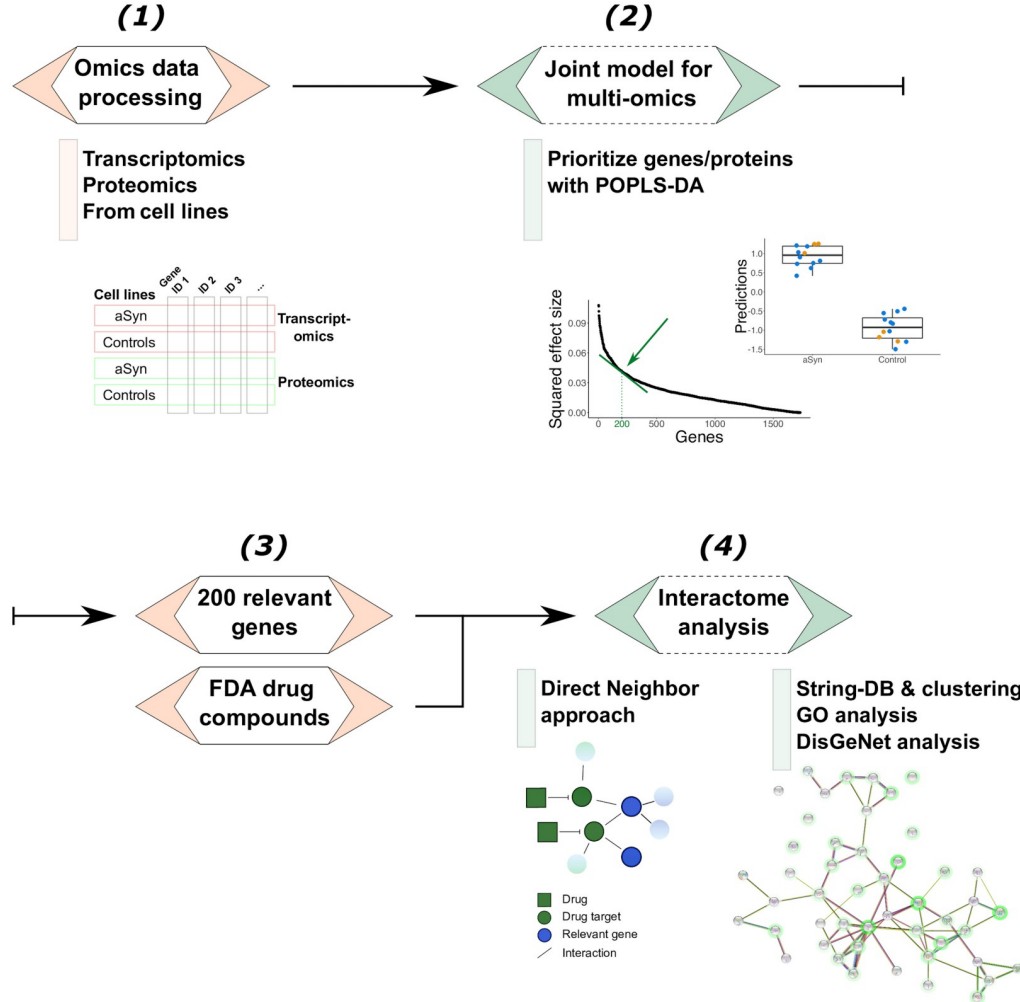

**Fig 1. Workflow of the multi-omics integration approach.** (1) To link the two datasets, the transcripts and proteins were mapped to their Entrez identifier, referred to as genes/proteins. (2) To identify the relevant genes/proteins that discriminate between the two experimental groups ($\alpha$-synuclein overexpression vs. control), POPLS-DA was applied. (3) The prioritized genes/proteins were integrated with the FDA-approved drug screening data using a direct neighbor approach. As an illustration, in the figure under "Direct Neighbor approach" are two drug target genes (green circles), which are neighbors of genes identified in (2) (blue circles). (4) Bioinformatics analyses: interactome, GO, and DisGeNet [13] enrichment analyses analyses. Protein-protein interaction networks were built using String-DB [14] where each node is a gene. The genes that were direct neighbors of a drug target were highlighted with a green halo, where its color intensity is proportional to the number of direct neighbors. Abbreviations: aSyn: $\alpha$-synuclein, GO: gene ontology, FDA: Food and Drug Administration.

## Validation of the protective efficacy of the FDA-approved drugs

Previously, we identified 53 FDA-approved drugs as being protective against $\alpha$Syn-induced toxicity in a primary drug screening [11]. Now, using a dose-response analysis, 41 of these 53 compounds were validated as being protective. The maximum effective concentrations appeared to range from 2.5 $\mu$M to 80 $\mu$M.

## POPLS-DA integration of transcriptome and proteome data

For the integrated analysis of the pre-processed transcriptomics and proteomics datasets, we used POPLS-DA. For each dataset, we used two joint components representing the relationship between the two datasets and two data-specific components. The number of components was determined by a scree plot of the eigenvalues of the row-wise concatenated datasets (S1 Supporting information). Based on a scree plot of the squared effect sizes per gene (given by the squared elements of $W\beta$), presented in Fig 2, 200 genes/proteins were retained for further analysis. A list of all genes/proteins and their weights are shown in S1 Supporting information.

Of the total variance in the transcriptomics and proteomics data, 21 and 18 percent, respectively, were explained by the integrative parts. Omics-specific variation accounted for around 27 percent of the total variation. The selected relevant genes/proteins appeared to be able to distinguish between the two experimental groups (Fig 2 and S1 Supporting information). We performed 400 permutations (see S1 Supporting information) of the experimental group labels to test the null hypothesis of no relation between the experimental groups and the genes/proteins. We found that, under the null hypothesis, the probability of achieving the same discrimination (100% accuracy) as observed in the data was 0.0025 (one in 400 permutations).

## Integrated interaction network based on multi-omics and drug data

Fig 3 shows the integrated interaction network constructed using String-DB. Enlarged copies of these figures are available in S1 Supporting information. The network had significantly more interactions than expected from an arbitrary subset of 200 genes. The network enrichment p-value was smaller than $10^{-15}$. From the 41 drugs that were validated to reduce $\alpha$-synuclein-induced toxicity [11], we found 27 targeting at least one gene in the integrated interaction network. The efficacy of the maximal effective dose of each of these compounds is shown in Table 1.

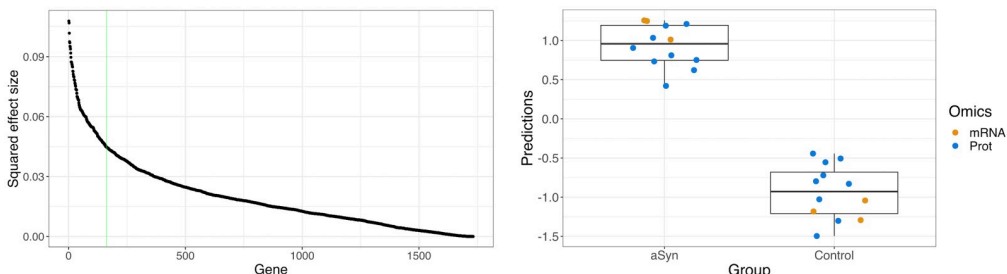

**Fig 2. Selection of relevant genes/proteins and their ability to discriminate.** The *left panel* shows the sorted squared effect size per gene given by the squared elements of $W\beta$. The 200 relevant genes/proteins correspond to the 'elbow', which is visually determined and approximately where the curve crosses the green vertical line. The *right panel* shows boxplots of score predictions based on the selected 200 relevant genes/proteins. A positive resp. negative prediction on the $y$-axis corresponds to a case resp. control. The dots, representing individual samples, are added with a horizontal 'jitter' to reduce overlap. The transcriptomics and proteomics samples are colored orange and blue, respectively. Abbreviations: aSyn: $\alpha$-synuclein case group, mRNA: transcriptome, Prot: proteome.

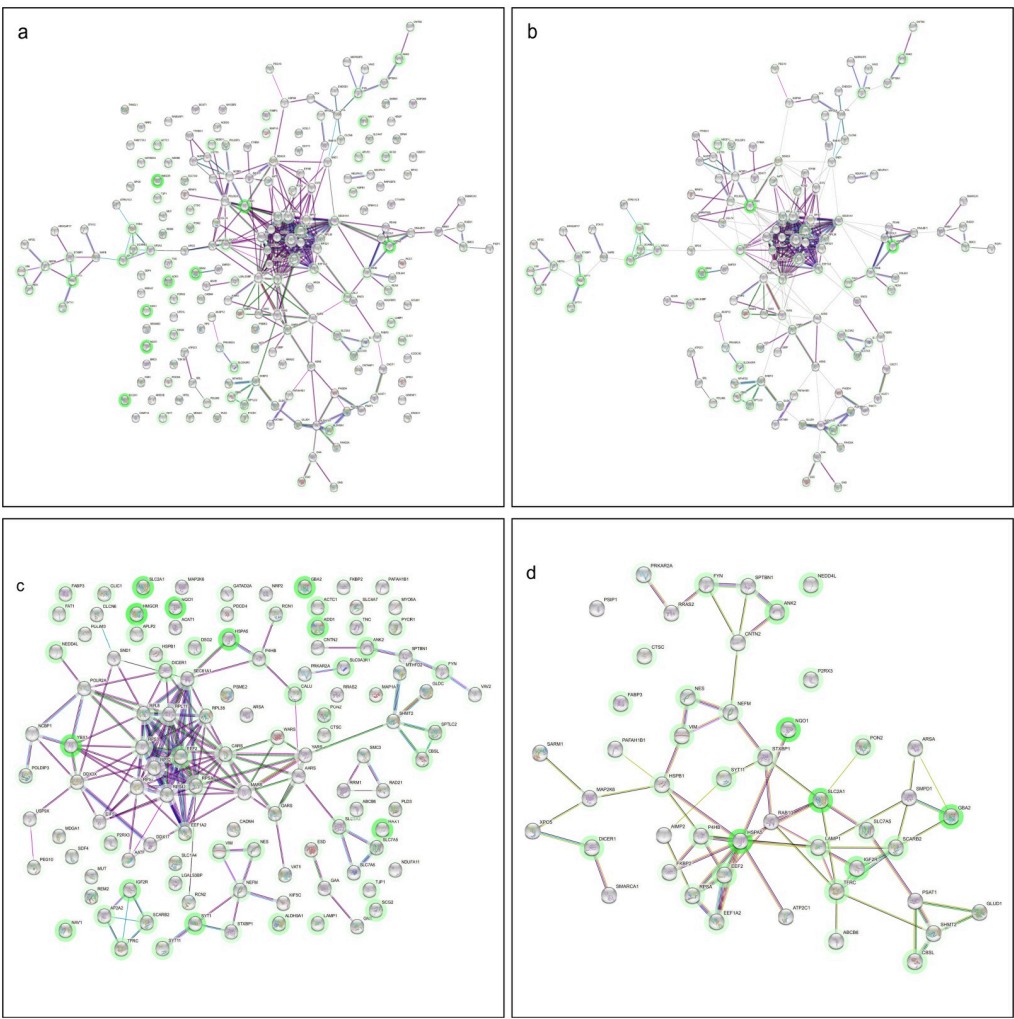

**Fig 3. String-DB and clustering analyses of the top 200 genes/proteins.** Figures are numbered from left to right, from top to bottom. Enlarged copies of these figures are available in S1 Supporting information. In panel (a), a network of interactions between the top 200 genes/proteins (estimated with POPLS-DA) was constructed using String-DB. Each node is a gene, and a connection between genes indicates evidence for a biologically plausible link. Text mining was excluded as an evidence source, and a medium confidence threshold was used. For genes that were (indirectly) targeted by a drug compound, a green 'halo' is drawn. The intensity of the green color is proportional to the number of drug compounds for which the gene was an (indirect) target. In panel (b), the interaction network from (a) was clustered using the MCL clustering algorithm from the String-DB website. The edges between the clusters are removed for visual aid. In panel (c), an interaction network is shown for a druggable subset of the top 200 genes/proteins, consisting of 116 genes that were (indirectly) targeted by an FDA-approved drug compound. In panel (d), an interaction network of the top genes in the "Parkinson's disease" DisGeNet term was constructed using String-DB. Text mining as evidence was included here.

The *HMGCR*, *NQO1*, *HSPA5*, and *YBX1* genes were most targeted by the validated drugs, namely by 20, 18, 17, and 17 drugs, respectively.

Table 1 shows for each compound the number of direct neighbors it targets in the network. Dexibuprofen, a non-steroidal anti-inflammatory drug (NSAID), targets the highest number of direct neighbors (N = 60), followed by Imatinib (N = 47), which acts on the tyrosine kinase enzyme and is involved in apoptosis. The most effective compound was Telmisartan with an

**Table 1. FDA-approved drugs protective against α-synuclein-induced toxicity with direct neighbors of the top 200 genes.** For each drug, the number of direct interactors is shown. The most effective concentration is the concentration at which each compound had the strongest protective effect. Cell survival is expressed as a percentage compared to αSyn-overexpressing cells that were treated with DMSO as control at the maximum effective concentration. P-Values of testing the dose-response were calculated with an ANOVA with Dunnett's post-hoc test for multiple comparisons.

| Drug name | Direct neighbors | Most effective concentration | Cell survival | P-value |
|---|---|---|---|---|
| Dexibuprofen | 60 | 80 $\mu$M | 118.29% | <0.0001 |
| Imatinib | 47 | 20 $\mu$M | 126.04% | <0.0001 |
| Amiodarone | 44 | 80 mM | 108.05% | 0.002 |
| Risperidone | 41 | 10 $\mu$M | 128.58% | <0.0001 |
| Bepridil | 36 | 80 $\mu$M | 115.89% | <0.0001 |
| Astemizole | 26 | 1.25 $\mu$M | 106.49% | 0.02 |
| Telmisartan | 24 | 5 $\mu$M | 133.16% | <0.0001 |
| Amlodipine | 21 | 10 $\mu$M | 116.42% | <0.0001 |
| Benazepril | 18 | 80 $\mu$M | 117.96% | <0.0001 |
| Repserpine | 18 | 80 $\mu$M | 107.64% | 0.01 |
| Dipyridamole | 16 | 40 $\mu$M | 118.09% | <0.0001 |
| Trifluopromazine | 16 | 40 $\mu$M | 114.30% | <0.0001 |
| Nefazodone | 15 | 40 $\mu$M | 115.08% | <0.0001 |
| Trimipramine | 15 | 20 $\mu$M | 125.54% | <0.0001 |
| Flunarizine | 14 | 20 $\mu$M | 104.38% | 0.03 |
| Quinacrine | 13 | 5 $\mu$M | 119.07% | 0.008 |
| Pentoxyverine | 10 | 1.25 $\mu$M | 109.69% | 0.002 |
| Dicyclomine | 8 | 20 $\mu$M | 120.15% | <0.0001 |
| Doxazosin | 8 | 40 $\mu$M | 116.12% | <0.0001 |
| Ajmaline | 7 | 20 $\mu$M | 108.94% | <0.0001 |
| Ifenprodil | 7 | 2.5 $\mu$M | 116.74% | 0.0003 |
| Lomerizine | 7 | 80 $\mu$M | 110.35% | <0.0001 |
| Pentamidine | 6 | 2.5 $\mu$M | 119.86% | 0.003 |
| Guanfacine | 5 | 5 $\mu$M | 117.01% | <0.0001 |
| Dyclonine | 4 | 5 $\mu$M | 125.69% | <0.0001 |
| Tropisetron | 3 | 20 $\mu$M | 131.63% | <0.0001 |
| Clemastine | 1 | 10 $\mu$M | 117.52% | 0.0004 |

improvement of the cell survival of α-synuclein-overexpressing cells by 33.16% at a concentration of 5 $\mu$M, followed by Risperidone which improved cell survival by 28.58% at 10 $\mu$M.

The network was split into sub-clusters using the MCL clustering algorithm on the integrated interaction network, see Fig 3. Several clusters consist of genes that are involved in neurogenerative disorders. The largest cluster is a highly connected hub of 22 genes, primarily ribosomal. Ribosomal proteins have been reported to mediate neurodegeneration in PD [15]. Another cluster of five genes includes the top two genes, *MTHFD2* and *CBS*. Overexpression of the *CBS* (cystathionine-beta-synthase) gene was protective in a PD toxin model in rats [16]. The network also contains clusters with genes involved in neurological development, e.g. *TFRC* (transferrin receptor), and neuronal adhesion and growth, e.g. *TNC* (tenascin C). Further, four genes encoding intermediate filaments, such as *NES* (nestin), are clustered together. They are primarily responsible for cell adhesion [17]. Furthermore, a cluster of three genes from the solute carrier family was found (*SLC3A2*, *SLC7A5*, and *SLC7A6*). These genes may play a role in the transport across the blood-brain barrier.

**Table 2. Gene ontology enrichment analysis of the top 200 genes.** Out of 1732 genes/proteins, the 200 genes/proteins with the highest weight (estimated using POPLS-DA) were used to perform GO enrichment analysis. All three GO ontologies were considered: cellular compound (CC), biological process (BP), and molecular function (MF). For each term, a p-value was computed using Fisher's exact test with the whole genome as background and corrected for multiple testing. The ten most significant terms are shown, ranked by their GO enrichment false discovery rate (FDR) adjusted p-value. The column "Genes in set" shows how many of the top 200 genes/proteins were annotated with the respective term.

| Term | Ontology | Genes in set | FDR |
|---|---|---|---|
| cytoplasm | CC | 178 | 1.28e-19 |
| extracellular exosome | CC | 71 | 4.67e-17 |
| extracellular vesicle | CC | 71 | 5.51e-16 |
| extracellular organelle | CC | 71 | 5.51e-16 |
| intracellular anatomical structure | CC | 190 | 7.62e-12 |
| cell junction | CC | 61 | 7.62e-12 |
| vesicle | CC | 88 | 2.30e-11 |
| extracellular space | CC | 81 | 7.09e-11 |
| protein binding | CC | 179 | 7.65e-11 |
| cytosol | MF | 100 | 1.48e-10 |

**Table 3. Top five disease enrichment clusters based on DisGeNet.** The top 200 relevant genes/proteins were analyzed using DisGeNet, a disease-gene database. The BEFREE database was used. For each term, a p-value was calculated using a Fisher exact test and FDR adjusted. The terms are ranked by this adjusted p-value.

| Term | Genes in set | FDR |
|---|---|---|
| Neoplasms | 142 | 2.84e-07 |
| **Parkinson's Disease** | 48 | 1.94e-06 |
| Carcinogenesis | 100 | 3.36e-06 |
| Malignant neoplasm of prostate | 75 | 2.54e-05 |
| Neoplasm Metastasis | 98 | 2.89e-05 |

## Enrichment analyses of relevant genes and proteins

In Table 2, the most significant pathways enriched in the top 200 genes are shown. The GO terms that contained "extracellular" and "cytoplasm" were the most significant. Genes with these annotations are involved in vesicular transport and signaling [18]. Nine of the ten terms were from the "cellular compound" GO category.

Table 3 shows the five diseases for which the most significant enrichment was found in the top 200 genes. The second term was "Parkinson's disease" and contained 48 genes. Fig 3 shows the PPI network for genes/proteins in this disease term. Note that the *HSPA5* gene, which was one of the most targeted by the FDA drug compounds, is within this network.

Finally, enrichment analyses were performed for the subset of 116 genes/proteins that were direct neighbors of genes targeted by at least one of the drugs. Fig 3 shows the protein-protein interaction network for this subset. The corresponding network enrichment p-value was smaller than $10^{-10}$. Also, the most significant GO terms agreed with the GO analysis of the total set of 200 genes/proteins (S1 Supporting information).

## Comparison of POPLS-DA with other omics integration approaches

Integrative multi-omics MINT, the single omics multivariate LASSO penalized regression and the single omics univariate t-testing are used as alternative approaches to analyzing the

**Table 4. Comparison of PD+MSA enrichment for single- and multi-omics approaches.** For multi-omics (POPLS-DA and MINT) and single-omics (LASSO and t-test) approaches, the selected genes are tested for DisGeNet Parkinson's disease and MSA enrichment. The FDR-corrected enrichment p-value is given in the second column. In the third column, the number of genes within the PD+MSA gene set out of the total number of selected genes is given. POPLS-DA and MINT were applied integratively (in bold italics) and individually to transcriptomics and proteomics data. The LASSO and t-test approaches were applied to transcriptomics and proteomics individually. Then, the selected genes and proteins were intersected (in italics).

| Approach | | Enrichment p-value | Genes in PD+MSA |
|---|---|---|---|
| POPLS-DA | transcriptomics | 2.15e-2 | 41/200 |
| | proteomics | 8.85e-3 | 42/200 |
| | *integrative* | 2.41e-5 | 48/200 |
| MINT | transcriptomics | 8.85e-3 | 42/200 |
| | proteomics | 2.15e-2 | 41/200 |
| | *integrative* | 1.38e-3 | 44/200 |
| LASSO | transcriptomics | 1.00 | 31/200 |
| | proteomics | 1.00 | 31/200 |
| | *intersection* | 1.00 | 17/116 |
| t-test | transcriptomics | 1.00 | 4/21 |
| | proteomics | 4.26e-4 | 17/44 |
| | *intersection* | 1.00 | 0/2 |

transcriptomics and proteomics data. For comparison, MINT and POPLS-DA are also applied to the single omics datasets individually.

Just as POPLS-DA, these approaches yielded high accuracies, namely all had a training accuracy of one. The permutation results showed that, except for LASSO, all methods had at most 4 instances where the accuracy was 1. Specifically, MINT and t-test approaches had 1 and 4 out of 400 instances of perfect accuracy, respectively. For LASSO, all 400 permutation instances yielded an accuracy of 1. POPLS-DA had perfect accuracy in 1 out of 400 permutations.

The results of the DisGeNet Parkinson's disease and MSA (PD+MSA) enrichment analyses of the obtained gene sets from the different methods are given in Table 4. The single-omics results are based on the transcriptomics and proteomics data individually, taking the 2292 overlapping genes and proteins.

POPLS-DA and MINT yielded significant enrichment of the obtained gene sets from single omic analyses as well as the integrative analysis. Enrichment of the gene set obtained from t-tests on proteomic data was also significant. None of the other single-omics approaches yielded a gene set significantly enriched for PD+MSA. For POPLS-DA on transcriptomics and proteomics data individually, we found 41 and 42 proteins in the PD+MSA gene set, respectively. The intersection of these gene sets with the 48 genes and proteins from the original integrative POPLS-DA multi-omics approach was 19 resp. 29 proteins.

## Discussion

In this study, we proposed a computational workflow to integrate omics and drug screen data from cell lines. A novel method, POPLS-DA, for the analysis of multiple omics datasets in non-overlapping cell lines was developed. The workflow is illustrated by jointly analyzing multi-omics and drug screening data from LUHMES cells overexpressing $\alpha$-synuclein and those overexpressing GFP as a control protein. Transcriptomics and proteomics data were first integrated with POPLS-DA. We found 200 relevant genes/proteins based on an elbow plot of the squared effect size of each gene (Fig 2). These genes/proteins appeared to perfectly discriminate between the two experimental groups.

Using a dose-response approach, 41 FDA-approved drugs were validated to be protective against $\alpha$-synuclein-induced toxicity. An integrated interaction network was constructed based on the 200 relevant genes/proteins by identifying direct neighbors of the 41 drugs in the network. Interactome analyses revealed several genes and pathways linked to PD, the most common synucleinopathy.

The drugs that targeted the highest number of direct neighbors were Dexibuprofen and Imatinib. Both drugs were previously found to lower the risk of PD in epidemiological studies [19, 20]. Interestingly, independent of the screening, we previously investigated the neuroprotective potential of imatinib as a macroautophagy-stimulating drug. Despite the protective effect, imatinib in our model led to a reduction of the expression of the dopaminergic markers tyrosine hydroxylase (TH) and dopamine transport (DAT) [21]. Therefore, further investigation would be necessary before using this drug in PD patients. However, other drugs we identified could be readily investigated in PD. Telmisartan, an AT1-blocker that is used for arterial hypertension, was the most effective drug with an improvement of cell survival by >30% at a comparably low concentration of 5$\mu$M. Since there are no major restrictions in PD other than a blood-pressure-lowering effect, Telmisartan could be a promising candidate for neuroprotective therapy. Similarly, Benazepril, an ACE inhibitor and blood-pressure-lowering drug, could be investigated with similar caution. Interestingly, AT1-blockers and ACE inhibitors have been shown to reduce the risk of falls in PD, and this association was not related to blood pressure [22]. Furthermore, the use of AT1-blockers and ACE inhibitors as potential neuroprotective drugs was previously discussed, because these drugs were protective in PD animal models, most likely due to an anti-oxidative effect [23]. In line with that, our data further emphasize that it would be promising to further investigate AT1-blockers and ACE-inhibitors in PD.

Other drugs, including Risperidone and Trifluopromazine, are antipsychotic drugs with an anti-dopaminergic effect. Though validated to be protective, they can lead to drug-induced parkinsonism [24] and are contraindicated in PD. Also, Flunarizine, a drug used in migraine prophylaxis can lead to drug-induced parkinsonism [25, 26].

The genes that were targeted by the majority of the drugs were *HMGCR*, *NQO1*, *HSPA5*, and *YBX1*. Note that *HMGCR* (3-Hydroxy-3-Methylglutaryl-CoA reductase) is the main target of statins. *NQO1* (NAD(P)H quinone dehydrogenase 1) has multiple functions and has been linked to Alzheimer's disease [27]. The *HSPA5* gene (heat shock protein family A (Hsp70) member 5) encodes a heat shock protein of the HSP70 family. It is found to be involved in maintaining the correct folding behavior of proteins [28]. HSPA5, also known as glucose-regulated protein 78 (GRP78), is located on the membrane of the endoplasmic reticulum and is essential for the unfolded protein response [29]. Furthermore, it was previously shown that overexpression of this protein reduced $\alpha$-synuclein toxicity to dopaminergic cells in a rat PD model [30]. Interestingly, a previous study showed that GRP78 levels were reduced in the temporal cortex and cingulate gyrus of PD patients compared to healthy controls but markedly increased in PD patients with dementia and patients with dementia with Lewy bodies [31]. In line with that, our data that show an interaction of drugs that protect against $\alpha$-Syn with this protein support that GRP78 plays an important role in $\alpha$-Syn toxicity and deliver further evidence that GRP78 could be a promising target for neuroprotective therapies of PD. The *YBX1* gene (Y-box binding protein 1) is a cold shock domain protein, predominantly localized to neurons and essential for brain development [32]. Furthermore, several clusters were observed in the network involving ribosomal proteins and intermediate filaments. Enrichment analyses identified several terms involving "extracellular" and "vesicles".

Using DisGeNet, the second most significantly enriched disease term was "Parkinson's disease", with 48 genes from our list of relevant genes/proteins. The other disease terms involved

neoplasms and carcinogenesis. Note that previous studies suggested shared molecular mechanisms between PD and cancer [33–35].

In a previous study on these data [1], the authors identified significant deregulation of synaptic proteins, in particular, *STXBP1*, *STX1B*, *SYT1*, and *FYN*. In our POPLS-DA integration, we found three of these genes: *STXBP1*, *SYT1*, and *FYN*, ranked 16, 18, and 72, respectively. We identified two additional synaptic genes in the top 200, namely *STX12*, encoding syntaxin-12, as well as *SYT11*, encoding synaptotagmin-11. Single nucleotide polymorphisms in *SYT11* have been linked to PD in a previously performed GWAS [36]. Furthermore, a previous study showed that synaptotagmin-11 plays a role in Parkin-related parkinsonism. Additionally, it was previously reported that lysosome-related proteins, *SCARB2*, *CTSB*, and *SMPD1*, were differentially regulated upon overexpression of $\alpha$-synuclein in LUHMES cells [1]. From our top genes/proteins, 22 were annotated with the lysosome GO term. Among these 22 genes are the previously found *SCARB2* at position 7 in our top genes/proteins, as well as *SLC7A5* and *SLC3A2* at positions 5 and 6, respectively. The authors pointed to lysosome membrane protein 2 expression (encoded by *SCARB2*) as a therapeutic target to increase lysosomal glucocerebrosidase activity to promote $\alpha$-synuclein clearance. Note that one of our clusters in Fig 3 consists of genes encoding lysosome-related proteins (i.e., *SCARB2*, *AP2A2*, *ATP6V1C1*, *IGF2R*, and *TFRC*) that are also direct neighbors of the FDA-approved drugs.

The POPLS-DA gene set was more significantly enriched for Parkinson's and MSA (PD +MSA) disease terms (p = 2.41e-5) than other single- and multi-omic approaches. Also, the discrimination between cases and controls was statistically significant, with 1 out of 400 permutations yielding the same accuracy of 1 as in the data set. Applying POPLS-DA to transcriptomics and proteomics data individually also provided a significant enrichment of PD+MSA terms, but less significant (p = 8.85e-3) resp. (p = 2.15e-2). In addition, the overlap with the selected genes from the integrative POPLS-DA approach was only 19 resp. 29 genes. Even though POPLS-DA applied to each dataset individually yielded a significant enrichment, an integrative approach appeared to result in a more interpretable set of genes compared to only proteomics data. Since the proteomic measurements were found to be noisier than the transcriptomic measurements, more proteomic samples were measured, which might be an explanation for more significant results in the proteomic dataset. On the other hand, with the integrative approaches, including transcriptomics data was clearly beneficial for the overall results.

The data used in this study were acquired from a human (LUHMES) cell line, where each dataset was measured in a different sample. Most data integration methods require multiple datasets measured in the same samples to calculate correlations across omics datasets from which joint components can be derived. These methods are sometimes referred to as horizontal integration approaches, where the data are laid out next to each other with the samples across the rows. In the present paper, we proposed a framework for vertical integration, where different groups of samples (representing different omics data) are laid down below each other with the same (type of) variables across the columns. An assumption here is that the effect of each variable has the same direction across the omics data. For example, the mRNA and protein with the same gene ID are assumed to be over- or underexpressed simultaneously. This assumption can be enforced by applying a sign correction to each variable. Since POPLS-DA components are linear combinations, the sign can be reverted after estimation to ensure a correct interpretation of the effect direction if desired. Currently, POPLS-DA is restricted to the case that no missing variables are present across each group of samples.

Several databases are available with information on the interaction between genes and other molecular variables (interactomes) [37]. We chose String-DB since it was found to perform well concerning disease prediction [38]. However, they also showed that other databases

might comprise complementary information. Currently, to the best of our knowledge, no method can utilize all information across different interactomes. Information bias is another unsolved problem when using these databases: less studied diseases have less coverage in these databases [39]. Since synucleinopathies are less common and less studied (compared to, e.g., neoplasms), the bioinformatics interpretation of our results is likely to be biased towards well-studied diseases. Indeed, in DisGeNet, PD has only 2078 annotated genes, while neoplasms have more than 10000 genes. This could explain why our enrichment analysis based on DisGe-Net (Table 3) showed many genes annotated with "neoplasms".

The POPLS-DA method can integrate more than two datasets to distinguish between experimental conditions in cell lines if the variables in these datasets can be mapped to the same unit, i.e. genes. For example, SNPs from a GWAS study can be integrated by mapping them to genes. Here, the distance of a SNP to a gene or its transcription factor binding site [40] can be used. For methylation, CpG sites are commonly mapped to genes based on their distance to a transcription start site [41]. Since, typically, multiple SNPs and CpG sites are mapped to the same gene, an aggregation step is needed to obtain one measurement per gene. For example, the first principal component of the SNPs or CpG sites corresponding to a gene can be taken as the final aggregated measurement.

The data from the FDA-approved drugs were integrated and projected onto the interaction network of the 200 relevant genes obtained from the transcriptomics and proteomics data analyses. An alternative approach is to simultaneously analyze the transcriptomics and proteomics data together with data on the validated compounds, i.e. drug-augmented data integration. It has been shown that including such information via penalties into data integration analysis yields robust results [42]. Such a joint model can pinpoint genes that can discriminate the two groups while targeted by the compounds, improving the clinical relevance of the list of top genes.

## Conclusion

We validated 41 FDA-approved drugs to be protective against $\alpha$-synuclein-induced toxicity in dose-response analyses. We applied a novel data integration approach to combine these drug data with experimental multi-omics data. Our novel omics integration POPLS-DA identified a set of 200 relevant genes/proteins that discriminated between samples overexpressing $\alpha$-synuclein and controls, as well as validated drugs targeting these genes/proteins. The set of 200 genes was found to be significantly enriched for PD and MSA genes. Also, this enrichment was larger than the enrichment of gene sets obtained using other single- and multi-omics approaches. Some of the drugs (e.g. an ACE inhibitor and an AT1 blocker) could be readily investigated in PD patients. These findings can potentially be used to develop therapeutic targets for Parkinson's disease, multiple system atrophy, and other synucleinopathies.

## Materials and methods

### Processing samples and measuring data

We describe the screening of 1600 FDA-approved drugs and the acquisition of proteome and transcriptome data in brief. Details are available elsewhere [1, 11].

**FDA-approved drug compound validation.** For the dose-response testing, LUHMES cells were plated in flasks coated with poly-L-lysine (0.1 mg/ml) and fibronectin (5 $\mu$g/ml) in growth medium (DMEM/F12 with 1% N2 supplement and 0.04 $\mu$l/ml basic fibroblast growth factor). After 24h, the medium was changed to differentiation medium (DMEM/F12 with 1% N2-supplement, 1 $\mu$g/ml tetracycline, 0.49 $\mu$g/ml N6,2'-O-dibutyryladenosine 3',5'-cyclic monophosphate, 2 ng/ml glial-derived neurotrophic factor). After another 24

hours, the cells were transduced with adenoviral vectors leading to overexpression of human wild-type $\alpha$-synuclein. On the day after transduction, viral vectors were washed away, and the cells were replated on 384-well multi-well plates and treated with the different drugs solved in dimethyl sulfoxide. For validation, compounds were tested in concentrations between 0.6 nM to 20 $\mu$M. Compounds that were not protective or did not reach a protective plateau were further tested in higher concentrations up to 80 $\mu$M. The dose-response data were analyzed using an ANOVA with Dunnett's post hoc test to correct for multiple comparisons with GraphPad Prism version 9.4 for Windows 64-bit (GraphPad Software, San Diego, CA, USA). Fig 4 shows a schematic overview and results of the dose-response testing.

**Transcriptome and proteome data acquisition and processing.**    For the transcriptome and proteome data, cells were plated directly in differentiation medium and transduced with adenoviral vectors leading to overexpression of either human wild-type $\alpha$-synuclein or GFP, respectively. After 24 hours, the remaining viral vectors were removed. Samples were collected four days after transduction. Transcriptome data were measured with Illumina HumanHT-12_V3 bead chips (Illumina, San Diego, CA, USA). Proteome data were measured by liquid chromatography-mass spectrometry (LC-MS/MS). The transcription probes were filtered based on the Illumina detection p-value, where probes were removed when all samples had a p-value above 0.05. A variance filter was applied where probes were removed if the inter-quantile range (IQR) of a probe did not exceed the 0.5-th quantile of all IQRs. Probes that could not be mapped to an Entrez identifier were removed.

After pre-processing and filtering, the transcriptomics and proteomics datasets contained 15660 transcription probes and 2577 proteins that could be mapped to Entrez identifiers. Of these identifiers, 2292 were present in both datasets. We excluded the *SNCA* measurements from the data analysis.

For the transcriptomics data, we analyzed three samples from LUHMES cells overexpressing $\alpha$-synuclein, and three samples from LUHMES cells expressing GFP as a control protein. For proteomics, the numbers were equal to nine. We determined the sign of the t-statistic of each mRNA and protein separately with respect to the case-control grouping. When the sign of the corresponding protein's t-statistic differed from that of the mRNA, the measurements for that protein were multiplied by minus one. This adjustment ensures that the difference in means between cases and controls had a consistent sign across all omics data. The measurements for both datasets were scaled to have zero mean and unit variance.

## Computational workflow for multi-modal data

Our workflow for integrated analysis is depicted in Fig 1. To summarize, the workflow begins with preprocessing the multi-omics data into a data matrix where variables are mapped to the same nomenclature. We then propose a joint model, POPLS-DA, to analyze the multi-omics data in terms of the experimental groups.

Next, we prioritize the most relevant genes based on POPLS-DA and integrate them with validated drugs using a 'direct neighbor' approach. This step involves identifying drugs that significantly reduce the toxicity of $\alpha$-synuclein aggregation and retrieving the list of genes targeted by these drugs from DrugBank. We then add their direct neighbors in the String-DB database to the gene lists for each compound.

Finally, we construct an integrated gene-gene interaction network based on the relevant genes and perform functional enrichment analysis. The network is built with String-DB, and the nodes are colored according to the number of compounds targeting each gene. We use the Markov cluster algorithm to identify sub-clusters of genes/proteins. Gene ontology pathway

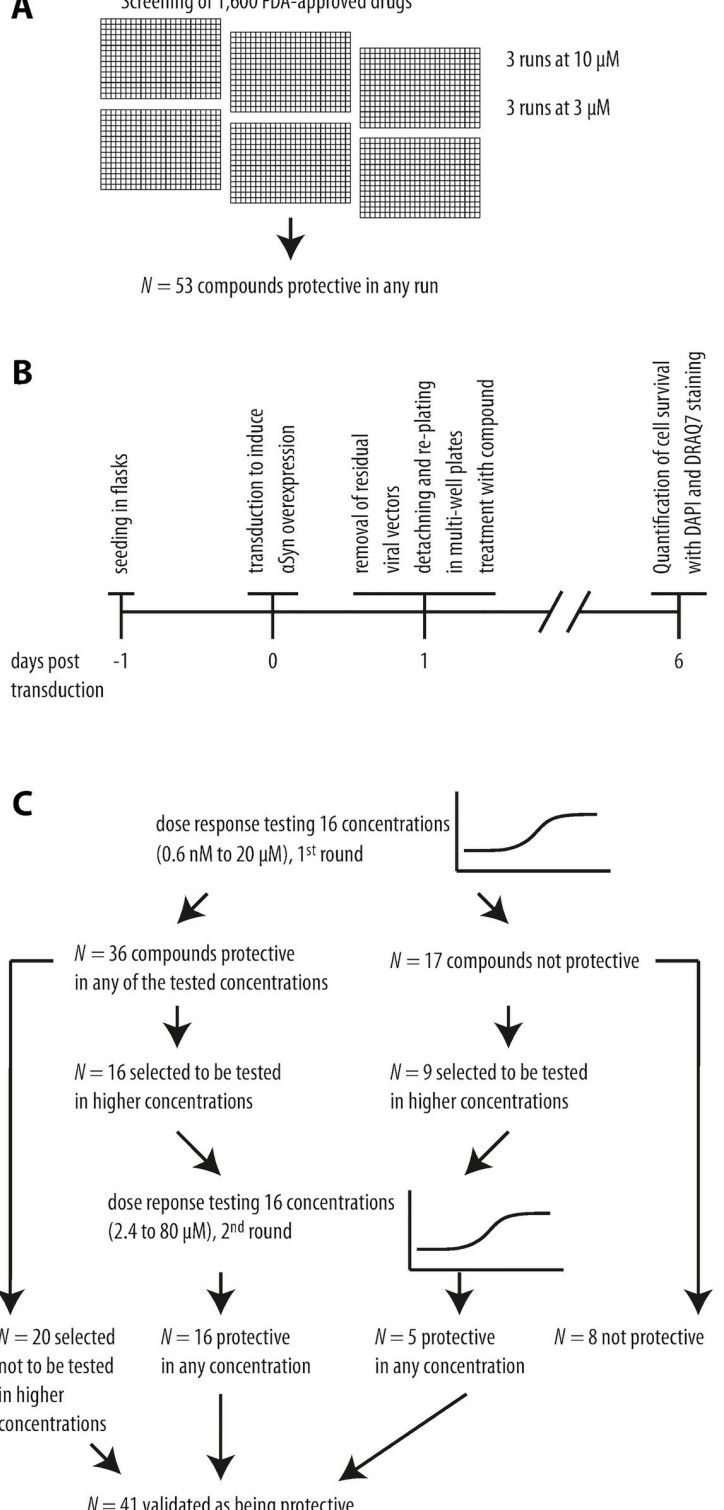

**Fig 4. Workflow of FDA-approved drug screening and dose-response validation.** 1,600 FDA-approved drugs were screened in two different concentrations (3 $\mu$M, 10 $\mu$M) in three runs each in an $\alpha$Syn toxicity cell model. 53 compounds were identified as being protective in at least one run (A). Timeline of dose-response testing (B). In the first round, concentrations from 0.6 nM to 20 $\mu$M of each compound were investigated. 36 compounds could be confirmed as being protective. Of these, 20 already reached their maximal protective potential and were selected not to

be tested in a higher concentration. The remaining 16 compounds were tested again in concentrations between 2.4 and 80 $\mu$M. All of these were confirmed as being protective in at least one concentration. Of the 17 compounds that were not protective in the first round of dose-response testing, 8 were already toxic and considered not protective. The remaining 9 were tested again in concentration from 2.4 to 80 $\mu$M. 5 of these compounds were protective in higher concentrations. In total, 41 could be confirmed as protective against $\alpha$Syn induced toxicity (C).

and disease enrichment analyses are performed using the `goana` function from the `limma` package in R and DisGeNet, respectively.

**POPLS-DA model for multi-omics cell line data.** For the joint analysis of the transcriptomics and proteomics datasets, a novel statistical data integration method, probabilistic orthogonal partial least squares discriminant analysis (POPLS-DA) is developed. Details can be found in S1 Supporting information.

Let $x_k$ be a random vector of size $p$ for $k = 1, \ldots, K$. In our data example, $k = 1, 2$ represents the 2292 transcriptomics and proteomics data, respectively. Similarly, let $y_k$ be a univariate random variable underlying the outcome. In our data example, $y_k$ are the two experimental conditions (one for cells overexpressing $\alpha$-synuclein, and zero for the controls) for $k = 1, 2$. In the POPLS-DA model, the group status and omics data are linked via latent variables $u_k$ of size $r$, with $r$ much smaller than the data dimensions. These latent variables represent joint parts of the two omics datasets that are correlated with the group status. To model omics variables that contribute to the variation in the omics dataset but do not play a role in distinguishing between the two experimental groups, specific components $v_k$ were added to the model. Residual variation is modeled by noise terms $e_k$ and $\epsilon_k$. All random variables are assumed to be zero-mean normally distributed. The mathematical model for POPLS-DA can be written as

$$x_k = u_k W^{\mathrm{T}} + v_k P_t^{\mathrm{T}} + e_k, \qquad y_k = u_k \beta + \epsilon_k, \qquad k = 1, \ldots, K. \tag{1}$$

The matrix $W$ contains the joint loadings, and the matrices $P_k$ contain the specific loadings for each gene in each component. The vector $\beta$ contains the regression coefficients of $y_k$ on the joint components $u_k$. The vector $W\beta$ represents effect sizes for the association of each variable with the outcome.

The observed datasets are assumed to consist of $N_k$ i.i.d. samples from the POPLS-DA model (1), with $k = 1, \ldots, K$. Note that the sample size per $k$ is allowed to differ. Based on these data, the parameters of POPLS-DA are estimated with memory-efficient maximum likelihood. Technical details are given in S1 Supporting information.

The numbers of components here are denoted by $r_k$, respectively. The number of joint and specific components, $r$ and $r_k$, have to be specified and can be based on scree plots [43] of the eigenvalues of the datasets. To select a set of relevant genes/proteins, the obtained effect sizes $W\beta$ are ranked and plotted. The elbow criterion [43] is used to determine the threshold between relevant and irrelevant genes/proteins.

**Computational workflow for integrating multi-omics and drug screening data.** Based on the POPLS-DA model (1) for multiple omics datasets, we propose a computational procedure for integration and analysis of cell line data. First, the multi-omics data are analyzed with POPLS-DA to prioritize genes that discriminate cases from controls. The relevant genes/proteins identified by POPLS-DA are further analyzed using several databases on protein-protein interactions (PPI), drug targets, gene ontology, and gene-disease association.

To construct an integrated interaction network of the relevant genes/proteins, we employed the String-DB [14] website and the `STRINGdb` package in R [44]. The network was based on curated, computational, and experimental data sources, text mining was excluded. A medium evidence threshold (0.4) was used to define an interaction (edge in the network).

To integrate the FDA-approved drug screening data with the obtained integrated interaction network, we first identified drugs that significantly reduced the toxicity of $\alpha$-synuclein aggregation. We implemented an automated R procedure using DrugBank [45] to retrieve the list of genes targeted by these drugs. Using the `string_db$get_neighbors` function, the direct neighbors of these targeted genes in the String-DB database were added to the gene lists for each compound. Then, the number of compounds targeting each gene in the $\alpha$-Syn-PPI network was determined. Nodes (genes and proteins) in the network were colored according to this number using the "payload" function on the String-DB website.

To identify sub-clusters of genes/proteins in the integrated interaction network, the Markov cluster algorithm (MCL) [46] on the String-DB website was used. The inflation parameter was set to two.

Finally, gene ontology (GO) pathway and disease enrichment analyses were performed. For GO, the `goana` function from the `limma` package [47] in R was used, with the whole human genome as background. For disease enrichment analysis, DisGeNet [13], a disease-gene association database implemented in the `disgenet2r` R package, was used. Here, the "BEFREE" text mining database was used for the disease-gene enrichment score as it contains more annotations regarding MSA and PD.

Our computational workflow, including POPLS-DA, is available as an R markdown file automating most of the tasks detailed above. After the datasets are loaded into memory, the file can be executed and an html file is generated with results and visualizations. The workflow is available on github.com/selbouhaddani/MultiOmicsCompWorkflow.

**Comparing POPLS-DA with competing multi- and single-omic approaches.** Integrative (MINT), multivariate single-omics (LASSO), and univariate single omics methods (t-test) were considered as alternative approaches to analyzing the transcriptomics and proteomics data. For comparison, POPLS-DA and MINT were also applied to single transcriptomics and proteomics data individually taking the 2292 overlapping genes and proteins. For the single omics approaches, results were integrated by taking the intersection of the obtained single omic gene lists. For each method, the obtained gene lists were evaluated in terms of prediction accuracy and enrichment by the DisGeNet Parkinson's disease and MSA (PD+MSA) gene set.

For MINT, we used the `mint.splsda` function from the `mixOmics` R package [48] and set `keepx`, the number of variables to retain, to 200. For LASSO, we used the `glmnet` function from the eponymous R package [49] using the binomial family of distribution and `alpha` set to one. We set the penalty parameter such that 200 genes resp. proteins are retained. For the t-test, we used the `t.test` function in R to test for a difference of means between the two experimental groups per gene and protein separately. Here, we selected only the genes and proteins that were significant after FDR correction using the `p.adjust` function.

To assess whether the selected genes can be used to reliably discriminate the two experimental groups, we permuted the group labels and applied each approach. The selected genes were then used to classify the (permuted) group label and calculate whether the accuracy was 1. POPLS-DA, MINT, and LASSO are able to directly estimate class labels for each sample. For the t-test, we fitted a linear discriminant analysis with the significant genes or proteins as predictors. If no genes or proteins were significant, the accuracy was set to zero. The number of permutations was set to 400.

To test for significant enrichment among the selected variables, we used a hypergeometric test comparing the ratio of PD+MSA genes among the selected variables to the default DisGeNet background genome. The resulting p-value was FDR corrected with respect to the number of DisGeNet terms.

## Supporting information

**S1 Supporting information. Supplementary materials for POPLS-DA and data analysis.**
The mathematical details and proofs for POPLS-DA are given, as well as additional results and
figures for the data analysis.
(PDF)

## Author Contributions

**Conceptualization:** Said el Bouhaddani, Matthias Höllerhage, Hae-Won Uh, Jeanine Houwing-Duistermaat.

**Data curation:** Matthias Höllerhage, Claudia Moebius, Marc Bickle, Günter Höglinger.

**Formal analysis:** Said el Bouhaddani, Hae-Won Uh, Jeanine Houwing-Duistermaat.

**Funding acquisition:** Günter Höglinger, Jeanine Houwing-Duistermaat.

**Investigation:** Said el Bouhaddani, Matthias Höllerhage, Hae-Won Uh, Günter Höglinger.

**Methodology:** Said el Bouhaddani, Matthias Höllerhage, Hae-Won Uh, Jeanine Houwing-Duistermaat.

**Project administration:** Günter Höglinger, Jeanine Houwing-Duistermaat.

**Resources:** Matthias Höllerhage, Claudia Moebius, Marc Bickle, Günter Höglinger.

**Software:** Said el Bouhaddani.

**Supervision:** Hae-Won Uh, Günter Höglinger, Jeanine Houwing-Duistermaat.

**Validation:** Said el Bouhaddani, Matthias Höllerhage.

**Visualization:** Said el Bouhaddani, Matthias Höllerhage.

**Writing – original draft:** Said el Bouhaddani, Matthias Höllerhage, Jeanine Houwing-Duistermaat.

**Writing – review & editing:** Hae-Won Uh, Claudia Moebius, Marc Bickle, Günter Höglinger, Jeanine Houwing-Duistermaat.

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
