## [Decision Letter · Decision Letter 0]

10 Aug 2023

Dear dr. Said,

Thank you very much for submitting your manuscript "Statistical integration of multi-omics and drug screening data from cell lines" for consideration at PLOS Computational Biology.

As with all papers reviewed by the journal, your manuscript was reviewed by members of the editorial board and by several independent reviewers. In light of the reviews (below this email), we would like to invite the resubmission of a significantly-revised version that takes into account the reviewers' comments.

In addition to the comments by the authors, please extend the introduction and cite other methods that predict drug effects from cell lines. Position your own previous work and the new work with respect to other studies. Provide guidance for the readers, in how far the current problem cannot be addressed by other methods. Please also include a baseline comparison as suggested by one of the reviewers.

We cannot make any decision about publication until we have seen the revised manuscript and your response to the reviewers' comments. Your revised manuscript is also likely to be sent to reviewers for further evaluation.

Sincerely,

Marcel Holger Schulz, Ph.D.

Academic Editor

PLOS Computational Biology

Pedro Mendes

Section Editor

PLOS Computational Biology

In addition to the comments by the authors, please extend the introduction and cite other methods that predict drug effects from cell lines. Position your own previous work and the new work with respect to other studies. Provide guidance for the readers, in how far the current problem cannot be addressed by other methods. Please also include a baseline comparison as suggested by one of the reviewers.

Reviewer's Responses to Questions

**Comments to the Authors:**

Reviewer #1: The authors propose a computational pipeline to jointly analyze unpaired multi-omics data from cell lines. As a main part of the workflow, the authors introduce POPLS-DA, a probabilistic orthogonal partial least squares approach for discriminative analysis. For the main results, POPLS-DA was applied to a multi-omics dataset to identify a subset of relevant genes that help discriminate affected from non-affected (aSyn vs control) samples. Subsequently, the authors perform a gene set enrichment analysis on the signature of 200 genes, thereby uncovering significant gene sets / pathways of biological relevance for the dataset at hand. This review focuses mostly on the proposed POPLS-DA method, as it comprises the main source of novelty for the submission.

Major:

My main concern with the paper is the lack of comparison with other baselines. Indeed, the authors suggest that existing approaches assume paired samples, while their proposed method does not. In that case, one may come up with a simple baseline such as analyzing the data modalities independently using existing and widely accepted methods such as tree-based approaches that provide means to quantify the feature/gene importance, and combining the results in a post-hoc manner.

The authors also claim that the results shown in the paper can only be obtained when jointly analyzing the data, and not in a single-omic analysis. However, no evidence is provided to back up this claim.

Minor:

- Figure 1 has poor quality and it is quite difficult to read the text parts.

- In the supplementary material, the table numbering need to be fixed as it currently shows [??] in the reference.

- There are some typos across the manuscript, e.g. page 2: “…the drug screening data to highlight(s) which part…”

Reviewer #2: Summary

In their manuscript “Statistical integration of multi-omics and drug screening data from cell lines” el Bouhaddani and colleagues propose a workflow for multi-omics and drug screening data integration and apply this workflow to study synucleinopathies using transcriptomics, proteomics, and drug screening data on LUHMES cell lines (“cases” samples, overexpressing

-synuclein and “control” samples, overexpressing GFP). In particular the goal of the study is to find genes that distinguish cases and controls and at the same time are druggable.

Multi-modal integration is a common problem in computational biology. Authors developed an extension of PLS-DA method for multi-omics integration that works on non-overlapping samples. Also they developed a workflow for co-analyzing protein-protein interactions between identified genes (that distinguish cases and controls) and drug target information for the most protective drugs (identified in the drug screen). Overall the paper presents an interesting technical approach, but in applying this approach to the biological data in question there are a number of major issues that should be addressed.

Major issues

1. The authors integrate transcriptomics dataset with 15660 transcripts and proteomics dataset with 2577 proteins and in the combined set they get 2292 genes. It means only less than 15% of genes profiled with transcriptomics were used in the analysis. So, if one would use the full transcriptomics dataset (e.g. w/o integration with proteomics) the set of genes that would be identified by a PLS-DA model would be different and also all the downstream findings with respect to drugs - targets analysis. Limiting analysis only to the genes in overlap between two omics sets biases/limits biological findings.

One additional point here. Since all samples in the analysis belong to the LUHMES cell line I wonder whether the transcriptomics and proteomics samples can be actually considered as the same i.e. “overlapping” samples. If such consideration is possible one could also perform a “vertical” integration that would preserve all the information in both datasets.

2. The transcriptomics dataset consists of 6 samples, while the proteomics has 18 samples. It’s unclear how well the POPLS-DA deals with cases when the sizes of integrated sets are imbalanced. Would be interesting to see whether the top 200 genes identified by PLS-DA model applied to just proteomics data are different from the top 200 genes obtained in the paper from the integrated dataset.

3. Page 13, line 311:

“The correlation of the transcripts with the -synuclein overexpressing cells versus the control cells was taken as a reference; if this correlation was negative, the corresponding protein feature was multiplied by minus one.”

This sentence should be re-formulated so it’s clear for the reader what is correlated with what. Also it’s unclear why this per gene correlation is calculated, and what is the reason for the subsequent transformation of protein features.

Minor issues

Fig. 1: the resolution is very low, it’s difficult to read.

Fig. 2, right: although it’s clear that score of 1 means “case” and score of -1 means control, perhaps it should be clarified in the legend or in the main text.

Fig. 3: panel names are missing on the figure. Also bigger font for gene names would make this figure better readable.

Fig. 4: this is a really good and important figure, it could be moved to the front of the paper so the reader understands the whole workflow right from the beginning (I would also show the number of transcriptomics and proteomics samples in the panel (1)). And the current Fig. 1 could be moved to the materials and methods or to the supplement.

Supplementary materials, page 4, line 103: “??” symbols instead of Table numbers.

It would be good to have a visual summary / illustration for POPLS-DA method in addition to “mathematical model” and “interpretation” sections in the supplement.

**Have the authors made all data and (if applicable) computational code underlying the findings in their manuscript fully available?**

Reviewer #1: Yes

Reviewer #2: Yes

PLOS authors have the option to publish the peer review history of their article (what does this mean?). If published, this will include your full peer review and any attached files.

Reviewer #1: No

Reviewer #2: **Yes: **Roman Kurilov
---

## [Decision Letter · Decision Letter 1]

13 Nov 2023

Dear dr. Said,

Thank you very much for submitting your manuscript "Statistical integration of multi-omics and drug screening data from cell lines" for consideration at PLOS Computational Biology. As with all papers reviewed by the journal, your manuscript was reviewed by members of the editorial board and by several independent reviewers. The reviewers appreciated the attention to an important topic. Based on the reviews, we are likely to accept this manuscript for publication, providing that you modify the manuscript according to the review recommendations.

The reviewers are generally happy with the revised manuscript. Please address the minor points by Reviewer 2 before publication.

Sincerely,

Marcel Holger Schulz, Ph.D.

Academic Editor

PLOS Computational Biology

Pedro Mendes

Section Editor

PLOS Computational Biology

The reviewer are generally happy with the revised manuscript. Please address the minor points by Reviewer 2 before publication.

Reviewer's Responses to Questions

**Comments to the Authors:**

Reviewer #1: I thank the authors for addressing every point in my initial review. I believe the comparison with other baselines further demonstrates the relevance of their proposed method. Therefore, I lean towards accepting this paper.

Reviewer #2: Authors addressed my comments in the revision, however I've noticed a couple of other statements in the text that would be good to clarify. Also I have questions to the authors's response to my original point number 3.

Line 78: “This approach highlighted genes in the network that are associated with α-synuclein overexpression and targeted by validated drugs, and therefore are potential targets for a novel therapy for synucleinopathies”

If these discovered genes are already targeted by existing drugs why is there a need for a novel therapy targeting the same genes? Conversely if the goal is to discover new potential targets for new therapies why is it important to look at the overlap of genes discovered by multi-omics analysis and genes that are targets of existing protective drugs (and not just to look at all genes discovered by multi-omics analysis).

Line 320: “Our novel omics integration POPLS-DA identified a set of 200 relevant genes/proteins that discriminated between samples overexpressing α-synuclein and controls, as well as their drug targets”

It’s unclear to which part of the sentence “as well as their drug targets” belongs. Also, whose drug targets? Drugs can have drug targets, genes/proteins can *be* drug targets but they cannot *have* drug targets.

Line 366: “We determined the sign of the t-statistic of each gene and protein separately with respect to the case-control grouping. When the sign of the corresponding protein's t-statistic differed from that of the gene, the measurements for that protein were multiplied by minus one. This adjustment ensures that the difference in means between cases and controls had a consistent sign across all omics data.”

1) Since you are talking about expression / transcriptomics data instead of “gene” you should use “mRNA” or “gene expression level”.

2) It’s unclear to me why the difference in means between cases and controls (for a gene) must have the same sign in transcriptomics and in proteomics data.

3) I understand that such genes (where difference in means between cases and controls in transcriptomics and proteomics data has the different sign) would not be good predictors for case / control prediction, but I am not sure that it’s a justifiable data transformation – to just multiply by -1 a subset of genes in proteomics dataset.

**Have the authors made all data and (if applicable) computational code underlying the findings in their manuscript fully available?**

Reviewer #1: Yes

Reviewer #2: Yes

PLOS authors have the option to publish the peer review history of their article (what does this mean?). If published, this will include your full peer review and any attached files.

Reviewer #1: No

Reviewer #2: **Yes: **Roman Kurilov

Figure Files:

Data Requirements:

Reproducibility:

References:

---

## [Editor Report · Decision Letter 2]

8 Jan 2024

Dear dr. Said,

We are pleased to inform you that your manuscript 'Statistical integration of multi-omics and drug screening data from cell lines' has been provisionally accepted for publication in PLOS Computational Biology.

Best regards,

Marcel Holger Schulz, Ph.D.

Academic Editor

PLOS Computational Biology

Pedro Mendes

Section Editor

PLOS Computational Biology

Thank you for addressing the remaining points of reviewer 2.

---

## [Editor Report · Acceptance letter]

23 Jan 2024

PCOMPBIOL-D-23-01100R2 

Statistical integration of multi-omics and drug screening data from cell lines

Dear Dr el Bouhaddani,

I am pleased to inform you that your manuscript has been formally accepted for publication in PLOS Computational Biology. Your manuscript is now with our production department and you will be notified of the publication date in due course.

With kind regards,

Zsofia Freund
